# Facial Nerve Graft in Malignant Tumors: The Role of Facial Rehabilitation

**DOI:** 10.3390/jcm14030968

**Published:** 2025-02-03

**Authors:** Francesco Mattioli, Costanza Galloni, Chiara Alberti, Marco Bonali, Alfredo Lo Manto, Stella Baraldi, Roberto Tonelli, Federica Nizzoli, Elena Reggiani, Alice Barbazza, Carlotta Liberale, Marco Ferrari, Matteo Fermi, Matteo Alicandri-Ciufelli, Ignacio Javier Fernandez, Elisabetta Zanoletti, Piero Nicolai, Daniele Marchioni

**Affiliations:** 1Department of Otorhinolaryngology—Head and Neck Surgery, Azienda Ospedaliero—Universitaria Policlinico di Modena, University of Modena and Reggio Emilia, 41125 Modena, Italy; 2Department of Otorhinolaryngology, Ospedale Infermi, 47923 Rimini, Italy; 3Ph.D. Program, Clinical and Experimental Medicine, University of Modena and Reggio Emilia, 41125 Modena, Italy; 4Respiratory Disease Unit and Center for Rare Lung Disease, Department of Surgical and Medical Sciences, Azienda Ospedaliero—Universitaria Policlinico di Modena, University of Modena and Reggio Emilia, 41125 Modena, Italy; 5Department of Otolaryngology—Head and Neck Surgery, IRCCS Azienda Ospedaliero—Universitaria Di Bologna, 40138 Bologna, Italy; 6Department of Otorhinolaryngology, University of Verona, 37134 Verona, Italy; 7Otolaryngology Section, Department of Neuroscience (DNS), University of Padova, 35123 Padova, Italy; 8Department of Otorhinolaryngology, Ospedale Santa Maria delle Croci, 48121 Ravenna, Italy

**Keywords:** head and neck malignancy, facial nerve sacrifice, facial nerve grafting, post-operative radiotherapy, facial rehabilitation

## Abstract

**Background**: Oncological surgery of the parotid gland or of the temporal bone may require the contemporary sacrifice of the facial nerve (FN). In such cases, the immediate repair of the sacrificed FN is recommended. The aim of this study is to evaluate the impact of facial rehabilitation (FR) and, secondarily, of post-operative radiotherapy (PORT) on the FN outcome after FN sacrifice and reconstruction via cable graft. **Methods**: This is a multicentric retrospective study including patients affected by malignant tumors whose surgical excision required FN sacrifice and contextual FN reconstruction with a cable graft. Other FN reconstruction techniques were excluded. FN function was assessed using both House–Brackmann and Sunnybrook grading systems. **Results**: A total of 28 patients were included. Most of the patients underwent a total parotidectomy. The greater auricular nerve was the main donor for cable graft. FR and PORT were performed in 22 and 15 patients, respectively. In particular, 20 patients underwent neuro-muscular retraining (NMR). Patients who underwent FR had better FN outcomes compared to those who did not (*p* = 0.02 at 12 months and *p* = 0.0002 at 24 months). In contrast, there was no statistically significant difference between patients who underwent PORT and those who did not (*p* > 0.05). Pre-operative FN palsy is a risk factor of worse FN function outcomes after cable graft. **Conclusions**: Our study, even though it was limited to only 28 cases, may demonstrate that cable graft failure is not due to PORT, as widely believed among clinicians, but to the absence of a rehabilitation program. Moreover, we suggest that the key to obtaining the best possible FN function results after FN sacrifice is the association of a technically correct FN reconstruction with a proper and targeted FR.

## 1. Introduction

Oncological surgery of the parotid gland or of the temporal bone may require the contemporary sacrifice of the facial nerve (FN) due to the impossibility of dissecting it from the neoplasm during radical excision of the tumor.

In particular, some studies have demonstrated a direct correlation between the tumor diameter and the risk of facial nerve invasion, with a probability of invasion that sometimes reaches 80% [1,2]. In such cases, the facial nerve needs to be sacrificed for oncological purposes, and immediate reconstruction with an inter-positional nerve graft is indicated where possible.

The aesthetic and functional deficits created by the sacrifice of the FN, in fact, have negative effects on social functioning, the emotional sphere, general mental health, and the vitality of patients compared to the normal population [3].

The immediate repair of the sacrificed FN is recommended in order to minimize the sequelae of facial paralysis, trying to restore facial tone and symmetry at rest along with voluntary and spontaneous facial movements.

In many cases, direct anastomosis of the FN is not feasible due to the loss of part of its length that hinders a tension-free suture of the two nerve stumps.

When direct repair is not possible, the restoration of facial symmetry and mimicry can be accomplished through various techniques. Listed in decreasing order of effectiveness, these include cable nerve grafts, anastomosis to hypoglossal or masseteric nerves or to the contralateral facial nerve (cross face), dynamic musculofascial transpositions (temporalis or gracile muscles), static musculofascial transpositions, and static procedures (eyelid chain or canthoplasty) [4].

Among these techniques, cable graft allows the emotional function of facial muscles; therefore, it is often the first choice in FN reanimation when direct anastomosis is not possible. Different graft can be chosen when performing this surgery; generally, the great auricular nerve is preferred due to its proximity to the surgical site and the multiple ramification anatomy [5]. Moreover, this nerve is always detected during oncological surgery of the parotid gland, but it is very rarely damaged or invaded by the tumor.

An important aspect to be considered in FN restoration is the role of facial rehabilitation (FR) performed by a speech and language pathologist. The utmost importance of FR in idiopathic or post-traumatic FN palsy is well known [6]. On the contrary, there is no literature addressing its role after cable graft. In clinical practice, FR is based on various different techniques: facial massage, thermotherapy, and facial neuromuscular retraining (NMR) [7].

In particular, NMR consists of non-surgical therapy conducted by a facial therapist who instructs the patient in selective motor control strategies designed to facilitate symmetrical movements and inhibit synkinesis [8]. In the literature, it is shown that NMR may increase the functional outcome of FN and decrease abnormal movement patterns, even in cases of longstanding paralysis [9].

In the case of FN sacrifice due to oncological reasons, postoperative radiotherapy (PORT) is commonly required to improve disease local control and long-term survival. However, the effects of PORT on FN grafts’ functional outcomes are still controversial, and few studies investigating this issue are available [4,10,11,12,13,14,15,16].

The aim of this study is to evaluate the impact of FR and, secondarily, of PORT on FN outcome after FN sacrifice and reconstruction via cable graft.

## 2. Materials and Methods

### 2.1. Study Design

This is a multicentric retrospective study performed in 4 Italian tertiary academic centers (Azienda Ospedaliero, Universitaria of Modena; Azienda Ospedaliera Universitaria Integrata of Verona; Azienda Ospedale, Università of Padova; and IRCCS Azienda Ospedaliero, Universitaria of Bologna). The study was conducted in accordance with the Declaration of Helsinki, and the study protocol was approved by the Ethics Committee “Area Vasta Emilia Nord” (261/23, ID 6052).

### 2.2. Study Population

We included patients affected by malignant tumors, whose surgical radical excision required FN sacrifice between January 2010 and May 2022. In all cases, surgery was performed using facial nerve intraoperative monitoring called NIM (nerve monitoring system) that detects the stimulation of both the main trunk and the peripheric branches and gives back an acoustic response.

The inclusion criteria were as follows: contextual FN reconstruction with a cable graft; at least a 12-month follow-up after surgery; availability of data about FN function grading, expressed by a validated Facial Nerve Grading System (House–Brackmann [17] or Sunnybrook [18]) before surgery, after surgery, and during follow-up.

The exclusion criteria were as follows: cable grafting not performed simultaneously with tumor excision; FN reconstruction techniques different from cable graft (such as hypoglossal- or masseteric-facial nerve anastomosis) even if performed in combination with cable graft; static reconstruction techniques (such as eyelid weight, canthoplasty, or tensor fascia lata sling) performed simultaneously with cable graft; anastomosis performed with fibrin glue and not with sutures.

All FN grafts were made with either 8-0 or 9-0 sutures using microscopic magnification. Patients’ clinical charts were retrospectively reviewed. Pre- and post-operative FN function was assessed using both House–Brackmann and Sunnybrook grading systems.

### 2.3. Statistical Analysis

The effects of FR, PORT, and pre-operative FN palsy on HB grading and Sunnybrook values over time were analyzed using ANOVA and linear regression. Additionally, univariable and multivariable logistic regression models were employed to investigate the association between PORT, SLT, and pre-operative FN palsy and the presence of HB > 4 at 12 months. A *p*-value of <0.05 was considered significant. Statistical analyses were performed using SPSS version 25.0 with the PSMATCHING3 R Extension command (IBM Corp., Armonk, NY, USA) and GraphPad Prism version 8.0 (GraphPad Software, Inc., La Jolla, CA, USA), unless otherwise indicated.

## 3. Results

### 3.1. Study Population

A total of 28 patients were eligible for this study according to the inclusion and exclusion criteria. Among these, there were 19 men (68%) and 9 women (32%), with a median age at surgery of 63 years (range 21–94). Regarding the risk factors, three patients were diabetic (11%), twelve patients were previous or active smokers (43%), and four patients were active alcohol consumers (14%).

A certain degree of pre-operative FN palsy was present in 12 patients (42.9%) (Figure 1).

None of the patients presented complete FN palsy (HB VI). The mean duration of pre-operative FN palsy was 3.6 months.

Table 1 reports the surgical details of the FN sacrifice and reconstruction. The FN was mainly sacrificed at its trunk (*n* = 22, 78%), and the anastomosis site was located, in most cases, in the extra-temporal segment of the FN (*n* = 17, 61%). The greater auricular nerve was the main donor for the cable graft (*n* = 20, 71%).

Most of the patients presented an immediate post-operative FN palsy HB grade VI (*n* = 22, 79%) (Figure 1).

The FN function expressed according to the Sunnybrook Facial Grading System was available for only 17 patients, with a mean composite score of 14.7. In particular, the mean voluntary movement, resting symmetry, and synkinesis scores were 31.3, 16.6, and 0, respectively.

Squamous cell carcinoma was the most common histology (*n* = 10; 35%). The tumor histologies are reported in Table 2.

The majority of patients (*n* = 22, 79%) underwent facial rehabilitation (FR) after surgery. The mean time between surgery and FR was 22.3 days (range 10–90 days). Twenty patients (71%) underwent NMR as a single rehabilitation technique or as part of a combined rehabilitation strategy.

During FR, three patients (11%) developed synkinesis necessitating botox injections.

Four patients (14%) underwent post-operative static reconstruction techniques, namely two simultaneous eyelid weight and canthoplasty, one eyelid weight, and one tensor fascia lata sling. The mean time between surgery and static reconstruction technique was 32.3 months (range 13–60 months). However, static reconstruction techniques were performed after a follow-up period of at least 12 months for every patient.

Fifteen patients (54%) received PORT: one patient underwent proton therapy, while fourteen patients underwent Intensity Modulated Radiation Therapy. The mean time between surgery and PORT was 71.2 days (range 40–184 days). The median radiation dose was 60 Gy (range 36–75 Gy). Three patients of the 15 who received PORT underwent chemotherapy too.

The median follow-up period was 21 months (range 12–30 months).

Among the 23 patients with severe FN palsy (HB V-VI) at the first post-operative evaluation, several patients improved their FN function during follow-up. In particular, at 12 months, six patients (26%) reached a moderate FN palsy (HB IV), and three patients (13%) reach a mild FN palsy (HB II-III). After that, the patients with mild FN palsy stopped their follow-up, while three patients improved from HB V to HB IV, and five patients improved from HB IV to HB III at 24 months.

### 3.2. Factors Affecting FN Outcomes

The patients who underwent FR had better FN outcomes compared to those who did not. This difference was statistically significant (*p* = 0.001), as demonstrated by the linear regression of the FN outcome, expressed using the HB scale, over time in patients who underwent FR or did not (Figure 2a). The analysis of variance (ANOVA) confirmed that this difference was statistically significant (*p* = 0.02 at 12 months and *p* = 0.0002 at 24 months) (Figure 2b).

Considering only NMR, the difference in terms of the FN outcomes, expressed using the HB scale, between those who underwent this technique and those who did not was statistically significant at the linear regression (*p* = 0.04) (Figure 3a) and at the ANOVA at 24 months (*p* = 0.0002) (Figure 3b). In contrast, when the FN outcomes were expressed using the Sunnybrook scale, no statistically significant difference was present between the two groups of patients, both according to the linear regression (Figure 3c) and ANOVA (Figure 3d).

Regarding PORT, there was no statistically significant difference between patients who underwent it and those who did not (*p* > 0.05), as suggested by the linear regression of the FN outcome, expressed using the HB scale, over time in the two groups (Figure 4a). In particular, at 24 months, the two lines of the linear regression tended to converge. These data were confirmed by the ANOVA (Figure 4b).

For the FN outcome expressed using the Sunnybrook scale, the difference between the two groups was confirmed to be not significant (Figure 4c,d).

Finally, Figure 5a displays the linear regression of the FN outcome, expressed using the HB scale, over time in patients who had pre-operative FN palsy (HB > 1) and in patients who had normal FN function before surgery (HB = 1). The second group had a better FN outcome, and this difference was statically significant (*p* = 0.005). These data were confirmed by the ANOVA at 24 months (*p* = 0.04) (Figure 5b), while when the FN outcome was expressed using the Sunnybrook scale, the difference was not confirmed to be statically significant (Figure 5c,d).

The independent association between clinical variables and HB > 4 at 12 months is summarized in Table 3.

Patients who did not undergo FR had an extremely increased risk of moderate–severe FN palsy at 12 months (odds ratio (OR) = 18.7). This result was statistically significant not only at the univariable analysis (*p* = 0.02) but also at the multivariable one (*p* = 0.03, OR = 17.1).

Similarly, patients who did not undergo NMR had an increased risk of moderate–severe FN palsy at 12 months (OR = 3.67), but this risk was not statistically significant (*p* > 0.05).

On the contrary, PORT was not associated with an increased risk of moderate–severe FN palsy (HB > 4) at 12 months (OR = 0.55). However, this association was not statistically significant (*p* > 0.05).

Finally, having a degree of FN palsy before surgery increased the risk of moderate–severe FN palsy at 12 months by 5 times (OR = 5.0), and this result was almost statistically significant (*p* = 0.05).

## 4. Discussion

A good FN function restoration after cable graft cannot be achieved without a facial rehabilitation program performed by a speech and language pathologist. As demonstrated by our analysis, patients who did not undergo FR had an extremely increased risk of moderate–severe FN palsy at 12 months. This result was statistically significant not only at the univariable analysis but also at the multivariable one. Both linear regression and ANOVA confirmed that the difference in terms of FN function outcomes between patients who underwent FR and those who did not was statistically significant.

NMR is the cornerstone of FR. Once again, the difference in terms of FN outcomes between patients who underwent this FR technique and those who did not was statistically significant.

To the best of our knowledge, this is the first study assessing the role of FR, and NMR in particular, after FN sacrifice and cable grafting.

In the common imaginary, PORT has always been considered harmful for FN function recovery after FN reconstruction, including cable grafting. As a consequence, some authors have stated that dynamic or static procedures should be preferred in patients eligible for PORT whose FN has been sacrificed. Pillsbury and Fisch [10] demonstrated how PORT reduced the average post-operative FN function outcomes from 70% to 25% of normal movements in 19 patients who underwent cable grafting following PORT from 1969 to 1975, and this difference was statistically significant. However, it necessary to specify that Intensity Modulated Radiation Therapy (IMRT), the radiation therapy technique mostly used today, was introduced into clinical practice in the late 1980s [19]; the patients in the aforementioned study underwent PORT with older techniques, burdened by a greater toxicity.

Gidley et al., in a retrospective review including 39 patients who underwent FN primary repair or nerve grafting [12], demonstrated that PORT appeared not to prevent achieving good FN function after nerve repair. As a matter of fact, good FN function was achieved in 50% of patients who received PORT compared to 80% of patients who did not receive PORT.

Brown et al. suggested that there was no difference in functional outcome between irradiated and non-irradiated patients after cable graft, but the median time to the best FN function after surgery was longer in patients who underwent PORT, even though this result was not statistically significant [4]. In a study based on a rabbit model, Zhou et al. confirmed that the time required to attain the best FN function postoperatively might be slightly longer for irradiated patients [14].

The first systematic review about the effect of PORT on FN grafts was published by Kenny et al. in 2023 [16]. Twelve studies, with a total of 142 patients, were included. The analysis demonstrated that there was no significant difference in postoperative FN function between PORT and non-PORT patients among patients undergoing FN grafting or repair. Starting from this systematic review, we developed our study, trying to overcome its limitations. First of all, the studies included in the systematic review by Kenny were heterogeneous in methodology. In particular, both direct repair and cable grafting were accepted as possible FN reconstructive techniques. In contrast, we decided to apply extremely strict inclusion and exclusion criteria, thus creating as homogeneous a group of patients as possible. As a consequence, our study included only 28 patients, even though it was a multicentric study, collecting the main Italian tertiary academic centers where FN surgery is regularly performed. Moreover, it must be noted that the studies included in the systematic review by Kenny [16] also had few patients, when considered separately, from a minimum of 3 to a maximum of 25. Finally, unlike Kenny et al. [16], we stated the FN function outcomes using not only the HB scale but also the more accurate Sunnybrook scale, which is able to detect tiny, yet clinically significant, differences.

Our study confirms that there is no statistically significant difference between patients who underwent PORT and those who did not after FN sacrifice and cable graft, in accordance with the recent literature on this topic. Therefore, we strongly recommend, in the case of FN sacrifice, contextual cable grafting with a donor nerve, when direct nerve repair is not possible, even if it is likely that the patient will be subjected to PORT.

Finally, our study demonstrates that patients with a certain degree of FN palsy before surgery have worse FN outcomes after cable graft compared to patients with normal pre-operative FN function. This result is statistically significant both according to linear regression and ANOVA at 24 months and almost significant according to univariable analysis, in accordance with the literature [4,12,20,21]. Probably, if the nerve is already damaged before its sacrifice, it will be incapable of regenerating through anastomosis [22].

Our study has some limitations. First of all, it is a retrospective study, but it is almost impossible to set up a prospective randomized study for tumors requiring FN sacrifice because of ethical problems, since the therapy is almost standardized.

According to our experience, graft length cannot be considered a confounding factor after 12 months follow-up.

Relatively to the paucity of the included patients, it is necessary to specify that these tumors are extremely rare in the population. Moreover, as mentioned above, we chose to apply strict inclusion and exclusion criteria in order to obtain a homogeneous group of patients. We have decided to exclude, among others, patients who underwent static reconstruction techniques such as eyelid chain and canthoplasty to reduce the bias on the FN function outcome. However, we would like to specify that eyelid chain does not obstruct proper FN function restoration after cable graft due to levator palpebrae superiorioris muscle hypertrophy. For that reason, it should be performed simultaneously with cable graft in order to prevent exposure to keratopathy, and it is not necessary to remove it even when an FN palsy HB grade III is reached.

## 5. Conclusions

Facial nerve surgery is a surgical niche because of its delicate procedures and the paucity of patients who can benefit from it. Therefore, it is of paramount importance that these patients are referred to specialized centers with a multidisciplinary team composed by FN surgeons, head and neck oncologic surgeons, speech and language pathologists, radiotherapists, oncologists, and neurologists. Only in this way is it possible to guarantee that FN reconstruction is performed during oncological demolishing surgery and that FR is carried out alongside PORT, thus ensuring the contemporary treatment of both malignancy and FN palsy.

Our study, even though it collected data from only 28 cases, suggests that cable graft failure is not due to PORT, as widely believed among clinicians, but to the absence of a rehabilitation program. This leads us to think that FN reconstruction without an appropriate FR may be useless and that a likely key to obtaining the best possible FN function results after FN sacrifice is the association of a technically correct FN reconstruction with a proper and targeted FR. Giving the paucity of data about this issue in the literature, more research should be conducted with a larger sample of patients.

## Figures and Tables

**Figure 1 jcm-14-00968-f001:**
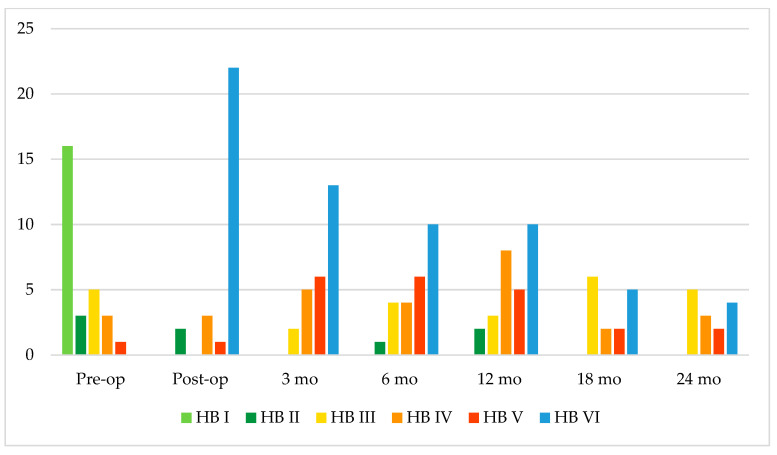
Pre-operative (*n* = 28), post-operative (*n* = 28), 3-month (*n* = 26), 6-month (*n* = 25), 12-month (*n* = 28), 18-month (*n* = 15), and 24-month (*n* = 14) follow-up facial nerve function according to House–Brackmann (HB) grading system.

**Figure 2 jcm-14-00968-f002:**
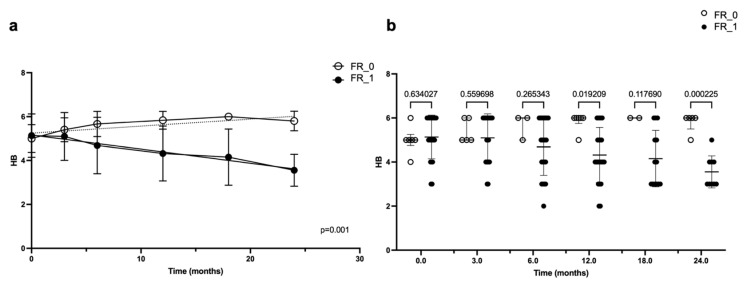
Linear regression (**a**) and ANOVA (**b**) of facial nerve outcome during time, according to House–Brackmann (HB) score, in patients who underwent facial rehabilitation (FR) and in those who did not. In both graphics, time is represented in months on the abscissa axes, while on the ordinate axes, the facial nerve function is indicated with the HB score; the *p*-value is reported in the bottom-right angle of the linear regression graphic, while in the ANOVA graphic, it is reported for each time interval (on the top of the figure).

**Figure 3 jcm-14-00968-f003:**
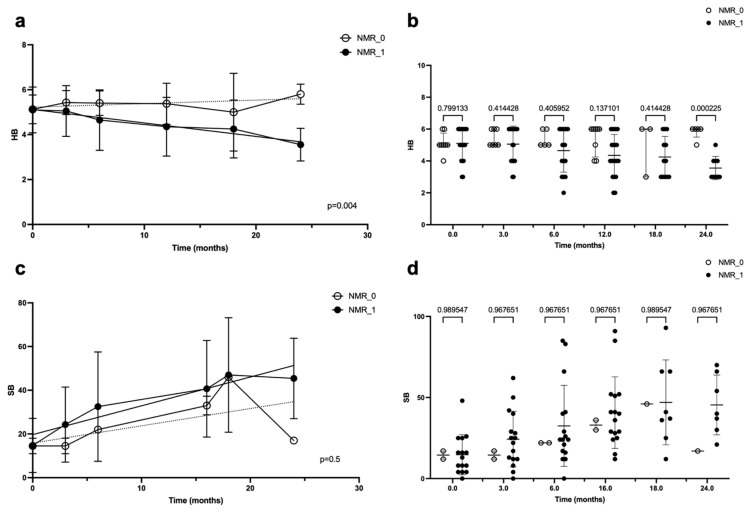
Linear regression (**a**,**c**) and ANOVA (**b**,**d**) of facial nerve outcome over time, according to the House–Brackmann (HB) score and Sunnybrook (SB) score, in patients who underwent neuro-muscular retraining (NMR) and in those who did not. In both graphics, time is represented in months on the abscissa axes, while on the ordinate axes, the facial nerve function is indicated with the HB score (**a**,**b**) or SB score (**c**,**d**); the *p*-value is reported in the bottom-right angle of the linear regression graphic, while in the ANOVA graphic, it is reported for each time interval (on the top of the figure).

**Figure 4 jcm-14-00968-f004:**
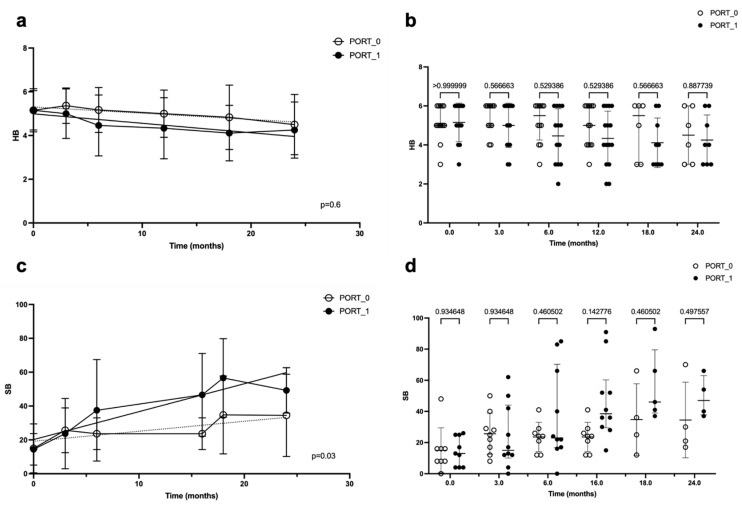
Linear regression (**a**,**c**) and ANOVA (**b**,**d**) of facial nerve outcome over time, according to the House–Brackmann (HB) score and Sunnybrook (SB) score, in patients who underwent post-operative radiotherapy (PORT) and in those who did not. In both graphics, time is represented in months on the abscissa axes, while on the ordinate axes, the facial nerve function is indicated with the HB score (**a**,**b**) or SB score (**c**,**d**); the *p*-value is reported in the bottom-right angle of the linear regression graphic, while in the ANOVA graphic, it is reported for each time interval (on the top of the figure).

**Figure 5 jcm-14-00968-f005:**
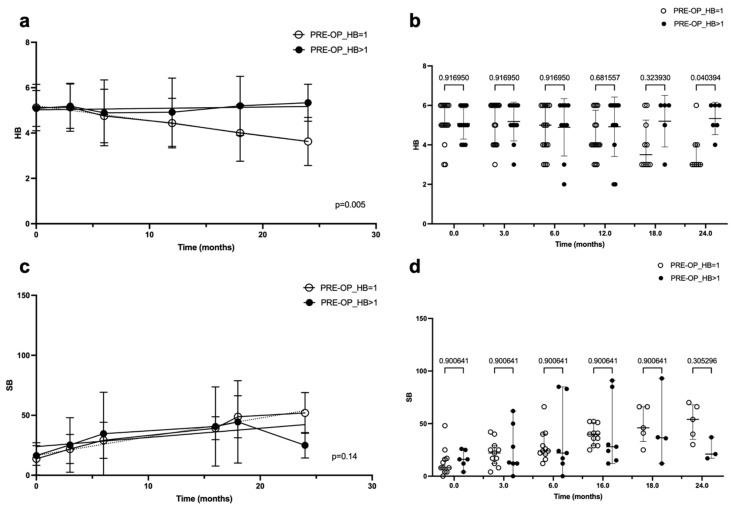
Linear regression (**a**,**c**) and ANOVA (**b**,**d**) of facial nerve outcome over time, according to House–Brackmann (HB) score and Sunnybrook (SB) score, in patients with a degree of pre-operative facial nerve palsy or not. In both graphics, time is represented in months on the abscissa axes, while on the ordinate axes, the facial nerve function is indicated with the HB score (**a**,**b**) or SB score (**c**,**d**); the *p*-value is reported in the bottom-right angle of the linear regression graphic, while in the ANOVA graphic, it is reported for each time interval (on the top of the figure).

**Table 1 jcm-14-00968-t001:** Surgical details about facial nerve sacrifice and reconstruction.

**Surgery**	***n* = 28 (100)**
Total parotidectomy	17 (60)
Total parotidectomy + mastoidectomy	8 (29)
Lateral temporal bone resection	3 (11)
**FN Sacrifice**	***n* = 28 (100)**
Main trunk	22 (78)
Temporo-facial trunk	4 (15)
Cervico-facial trunk	2 (7)
**Donor Nerves**	***n* = 28 (100)**
Greater auricular nerve	18 (64)
Sural nerve	5 (18)
Other nerve *	3 (11)
Greater auricular nerve + other nerve °	2 (7)
**Anastomosis Site**	***n* = 28 (100)**
Extra-temporal	17 (61)
Mastoid	11 (39)

*: accessory spinal nerve, medial antebrachial cutaneous nerve, lesser occipital nerve; °: ansa cervicalis, transverse cervical nerve.

**Table 2 jcm-14-00968-t002:** Tumor histologies.

Histology	*n* = 28 (100)
Squamous cell carcinoma	10 (35)
Adenoid cystic carcinoma	6 (21)
Adenocarcinoma	4 (14)
Salivary duct carcinoma	2 (7)
Melanoma	2 (7)
Mucoepidermoid carcinoma	1 (4)
Acinic cell carcinoma	1 (4)
Epithelial myoepithelial carcinoma	1 (4)
NUT carcinoma	1 (4)

**Table 3 jcm-14-00968-t003:** Independent association between clinical variables and HB > 4 at 12 months. FR: facial rehabilitation, NMR: neuromuscular retraining, PORT: post-operative radiotherapy, pre-op: pre-operative, OR: odds ratio, CI: confidence interval. *: Haldane–Anscombe correction.

	Univariable			Multivariable		
Parameter	OR	95% CI	*p* Value	OR	95% CI	*p* Value
**FR [0]** *	18.7	1.96–435	0.02	17.1	1.87–382	0.03
**NMR [0]**	3.67	0.66–29.5	0.02			
**PORT [1]**	0.55	0.11–2.32	0.43			
**Pre-op HB > 1 [1]**	5.0	1.03–30.1	0.05			

## Data Availability

The data presented in this study are available on request from the corresponding author due to privacy.

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
