# Peer review of "Facial Nerve Graft in Malignant Tumors: The Role of Facial Rehabilitation"

_jcm, 2025, doi:10.3390/jcm14030968_

Round 1
Reviewer 1 Report
Comments and Suggestions for Authors
Please address all the comments.
Abstract Results: Please specify the exact p-values in lines 36–39 to enhance the scientific rigor of the results section.
Abstract Conclusion: Given that this study included only 28 patients, drawing firm conclusions that contradict existing beliefs may be premature. Please revise the language to express less certainty and acknowledge the limitations of the study sample size.
Introduction: The introduction would benefit from mentioning ultrasonography of the facial nerve as a method to avoid nerve injury when possible.
Results: The inclusion of only 28 patients necessitates a recalculation of proportions with rounded percentages (e.g., replace "67.9%" with "68%"). Avoid decimal points in such small sample sizes, as they may imply unwarranted precision. This adjustment applies to all sections of the results.
Discussion: The discussion MUST include more comprehensive exploration of modern radiologic modalities that enable visualization of nerve fascicles and their application in improving fascicular alignment. Comment the recently published article in this field emloying modern radiological modalities for fascicular depiction (DOI: https://www.nature.com/articles/s41598-024-84396-y and discuss advancements such as high-resolution ultrasound and magnetic resonance microscopy, emphasizing their potential advantages in nerve fascicle depiction and alignment.
Conclusion: Similar to the abstract's conclusion, the main conclusion section should be reformulated to reflect the study's limitations and express less definitive statements, acknowledging the small sample size and the need for further research.
References: The reference list should be updated to include more recent literature, especially publications from the past five years, to reflect advancements in the field and support the study's findings with contemporary sources.
Reviewer 2 Report
Comments and Suggestions for Authors
The article by Francesco Mattioli and colleagues provides details of facial nerve graft in Malignant Tumors: the role of facial rehabilitation. The study primarily focused on investigating the impact of facial rehabilitation (FR) and postoperative radiotherapy (PORT) on FN outcomes after FN sacrifice and reconstruction via cable graft. The study consisted of a total of 28 patients out of 20 patients who underwent neuro-muscular retraining. Overall, this is a well-written article. The result shows that cable graft failure is not due to PORT, as widely believed among clinicians, but to the absence of a rehabilitation program.
The overall quality of the article must be improved, and the following changes should be made.
The introduction does not provide enough background information on facial nerve grafts in malignant tumors. The authors need to improve this section. The addition of figures is encouraged.
The authors need to increase the font sizes in Figure 2. Figures 2a and 2b should be represented on the same scale. The figure legends should include details of the statistical test.
Similar changes are also required in Figures 3, 4, and 5.
The Discussion section lacks specific literature references while discussing the study's results. The discussion section must include the relevant studies. E.g. from lines 269 to 287. The authors mention Kenny et al., on line 280, but the actual reference is missing while discussing the results.
